# Knockout of the Amino Acid Transporter SLC6A19 and Autoimmune Diabetes Incidence in Female Non-Obese Diabetic (NOD) Mice

**DOI:** 10.3390/metabo11100665

**Published:** 2021-09-29

**Authors:** Matthew F. Waters, Viviane Delghingaro-Augusto, Kiran Javed, Jane E. Dahlstrom, Gaetan Burgio, Stefan Bröer, Christopher J. Nolan

**Affiliations:** 1Australian National University Medical School, Australian National University, Acton, ACT 2601, Australia; matt.waters@anu.edu.au (M.F.W.); viviane.augusto@anu.edu.au (V.D.-A.); jane.dahlstrom@act.gov.au (J.E.D.); 2John Curtin School of Medical Research, Australian National University, Acton, ACT 2601, Australia; gaetan.burgio@anu.edu.au; 3Research School of Biology, Australian National University, Acton, ACT 2601, Australia; kiran.javed@anu.edu.au (K.J.); stefan.broeer@anu.edu.au (S.B.); 4ACT Pathology, The Canberra Hospital, Canberra Health Services, Garran, ACT 2605, Australia; 5Department of Endocrinology, The Canberra Hospital, Garran, ACT 2505, Australia

**Keywords:** amino acid, aminoaciduria, diabetes incidence, insulitis, neutral amino acid transporter, non-obese diabetic mouse, pancreatic islet, Slc6a19 deficiency, type 1 diabetes

## Abstract

High protein feeding has been shown to accelerate the development of type 1 diabetes in female non-obese diabetic (NOD) mice. Here, we investigated whether reducing systemic amino acid availability via knockout of the *Slc6a19* gene encoding the system B(0) neutral amino acid transporter AT1 would reduce the incidence or delay the onset of type 1 diabetes in female NOD mice. *Slc6a19* gene deficient NOD mice were generated using the CRISPR-Cas9 system which resulted in marked aminoaciduria. The incidence of diabetes by week 30 was 59.5% (22/37) and 69.0% (20/29) in NOD.*Slc6a19^+/+^* and NOD.*Slc6a19^−/−^* mice, respectively (hazard ratio 0.77, 95% confidence interval 0.41–1.42; Mantel-Cox log rank test: *p* = 0.37). The median survival time without diabetes was 28 and 25 weeks for NOD.*Slc6a19^+/+^* and NOD.*Slc6a19^−/−^* mice, respectively (ratio 1.1, 95% confidence interval 0.6–2.0). Histological analysis did not show differences in islet number or the degree of insulitis between wild type and Slc6a19 deficient NOD mice. We conclude that Slc6a19 deficiency does not prevent or delay the development of type 1 diabetes in female NOD mice.

## 1. Introduction

Islet beta cell metabolic stress in response to nutrient overload has been proposed to increase the susceptibility of islet beta cells to immune destruction, accelerating the development of type 1 diabetes in genetically at risk individuals [1,2]. Islet beta cell stress leads to activation of beta cell neoantigen production and stimulation of HLA class I expression on the cell surface [3,4,5]. This could underlie, at least in part, the increasing incidence of type 1 diabetes within older children and adolescents [6,7]. Accordingly, approaches to limit islet beta cell metabolic stress through pharmacological means, such as by using metformin, or reducing availability of nutrient secretagogues, could prevent type 1 diabetes [8,9]. Amino acids augment glucose-stimulated insulin secretion and have the potential to increase beta cell nutrient-induced stress [8]. There is pre-clinical evidence in situations of chronic beta cell stress, including endoplasmic reticulum stress, which shows transport of amino acids into beta cells is increased and this contributes to beta cell loss through apoptosis [10].

In human studies, in which the metabolome has been interrogated for biomarkers predictive of type 1 diabetes, trajectories of various amino acid plasma concentrations have been associated with the timing of islet antibody appearance [11,12,13]. In one study, elevated branched-chain amino acids (BCAA) and glutamine were shown to precede the development of insulin and glutamic acid decarboxylase antibodies in children who later progressed on to develop diabetes [13]. However, these trajectories are complex, differing according to which autoantibody appears first and at which age [11,12,13].

The non-obese diabetic (NOD) mouse spontaneously develops autoimmune diabetes at a high rate (80–90% in females) by 4–6 months of age [14]. The feeding of a protein-enriched diet to NOD mice was shown to accelerate the development of type 1 diabetes [15]. Furthermore, metabolomic analysis of plasma samples from NOD mice revealed elevated levels of BCAA in mice that did, compared to mice that did not, develop diabetes [16].

The *SLC6A19* gene encodes the broad-spectrum neutral amino acid transporter AT1 (B^0^AT1), the major epithelial transporter for neutral amino acids in the intestine and kidney [17]. Mutations in *SLC6A19* cause Hartnup disorder, which is mostly a benign condition in adults, characterised by aminoaciduria [17]. Global *Slc6a19* knockout on the C57BL/6J mouse caused marked aminoaciduria, reduced fed-state plasma amino acid levels, increased fibroblast growth factor 21 (FGF21), reduced glucose stimulated insulin secretion and improved glucose tolerance [18,19]. This suggests amino acid off-loading through *Slc6a19* deficiency reduces islet beta cell work, and consequently has the potential to reduce islet beta cell metabolic stress for prevention of both type 1 and type 2 diabetes [8,20].

The aim of this study was to determine if amino acid off-loading via *Slc6a19* knockout would reduce the incidence or delay the onset of type 1 diabetes in NOD mice.

## 2. Results

### 2.1. Female Slc6a19 Deficient NOD Mice Are Markedly Aminoaciduric

The CRISPR/Cas9 mouse construction of *Slc6a19* deficient NOD mice, as described in Materials and Methods, generated three mice with mutations in the *Slc6a19* gene, of which one (ASD707:Maggot:::F0#13) had a 7 bp deletion and a 2 bp insertion within exon 2 of a single allele. ASD707:Maggot:::F0#13 (NOD.*Slc6a19^+/−^* founder mouse) was chosen for this study. Female NOD.*Slc6a19^+/+^* and NOD.*Slc6a19**^−/−^* littermates were studied. The abundance in urine of alanine, glycine, isoleucine, leucine, methionine, phenylalanine, serine, threonine and valine were all markedly increased in 6–8 week old female NOD.*Slc6a19**^−/−^* compared to NOD.*Slc6a19^+/+^* mice (ranging from 90 fold increased for urinary leucine to 6000 fold increased for urinary methionine), consistent with successful construction of Slc6a19 deficient NOD mice (Figure 1a). While there was an overall trend for a reduction in plasma amino acid abundance in the NOD.*Slc6a19**^−/−^* compared to NOD.*Slc6a19^+/+^*, the differences were not significant (Figure 1b).

### 2.2. Diabetes Incidence Is Unchanged in Slc6a19 Deficient NOD Mice

The body weight of Slc6a19 deficient NOD mice was not different from that of the wild-type mice from baseline to study end (Figure 2a). NOD.*Slc6a19**^−/−^* compared to NOD.*Slc6a19^+/+^* mice, however, had mildly higher fed-state plasma glucose levels at 6 weeks of age (NOD.*Slc6a19^+/+^*, 5.6 ± 0.1 mmol/L; NOD.*Slc6a19**^−/−^*, 6.0 ± 0.2 mmol/L; mean ± SEM, *p* < 0.05), however, this difference did not track with increasing age (Figure 2b). Diabetes incidence by 30 weeks of age was 59.5% (22/37) and 69.0% (20/29) in NOD.*Slc6a19^+/+^* and NOD.*Slc6a19**^−/−^* mice, respectively (hazard ratio 0.77, 95% confidence interval 0.41–1.42; Mantel-Cox log rank test: *p* = 0.37) (Figure 2c). Diabetes developed as early as 14 weeks of age in two NOD.*Slc6a19**^−/−^*, however, the median survival time without diabetes was 28 and 25 weeks for NOD.*Slc6a19^+/+^* and NOD.*Slc6a19**^−/−^* mice, respectively (ratio 1.1, 95% confidence interval 0.6–2.0) (Figure 2c).

### 2.3. Insulitis Severity Was Unaltered by Slc6a19 Deficiency in NOD Mice

All 37 pancreases of the NOD.*Slc6a19^+/+^* mice and 26 of 29 pancreases of NOD.*Slc6a19^−/−^* were available for analysis. All available pancreases were used for analysis of islet number per pancreas section. As 3 pancreases of the NOD.*Slc6a19^+/+^* had no identifiable islets on the analysed section, these pancreases were excluded from scoring of insulitis. The number of islets per pancreas section did not differ between the mouse genotypes (NOD.*Slc6a19^+/+^*, 6 (4–9) islets/section; NOD.*Slc6a19**^−/−^*, 5 (4–9) islets per section; median (interquartile (IQ) range), *p* = 0.94) (Figure 2d). The severity of insulitis also did not differ with 19 of 34 (55.9%) and 11 of 26 (42.3%) of the NOD.*Slc6a19^+/+^* and NOD.*Slc6a19^−/−^* mice, respectively, having ≥50% of islets/pancreas section scoring at least grade 3 insulitis (chi-square test: *p* = 0.30). The average percentage of islets within each insulitis score category is shown in Figure 2e. As expected, the mice that developed diabetes, irrespective of mouse genotype, had a reduced number of islets per pancreas section (non-diabetic, 8.5 (4.5–15) islets/section; diabetic, 5 (3–6) islets/section; median (IQ range), *p* < 0.005) (Figure 2d) and a higher percentage of islets/pancreas section with at least grade 3 insulitis (non-diabetic, 7 of 22 (31.8%); diabetic, 23 of 38 (60.5%); *p* < 0.05).

## 3. Discussion

The major finding is that neutral amino acid off-loading by knockout of the amino acid transporter gene *Slc6a19* in female NOD mice did not delay or reduce the incidence of autoimmune diabetes development. Approximately two-thirds of the female mice developed diabetes by 30 weeks of age in both the NOD.*Slc6a19^+/+^* and NOD.*Slc6a19^−/−^* mice groups. Furthermore, there were no differences in the severity of insulitis between the two genotypes. Considering that a protein enriched diet has previously been shown to accelerate the onset of diabetes in female NOD mice [14], it might have been expected that reducing the amino acid load would at least delay the onset of diabetes, but this was not seen. Untested is whether the NOD.*Slc6a19^−/−^* would have prevented the acceleration of autoimmune diabetes if they were fed a high protein diet.

The phenotype of NOD global *Slc6a19* knockout mice in this study had similarities and differences to C57Bl6J global *Slc6a19* knockout mice [18,19]. *Slc6a19* knockout induced marked aminoaciduria in mice of both backgrounds, but growth restriction, with reduced birth weight and subsequent weight gain, was seen only in C57Bl6J.*Slc6a19^−/−^* mice [18]. Interestingly, fasting glucose tended to be slightly higher in C57Bl6J.*Slc6a19^−/−^* mice at two months of age, which is similar to the mildly elevated fed-state glucose levels at six weeks of age in NOD.*Slc6a19^−/−^* mice. There was a trend for lower plasma amino acid abundance in the NOD.*Slc6a19^−/−^* compared to NOD.*Slc6a19^+/+^* mice, with two of four mice of the NOD.*Slc6a19^+/+^* having much higher levels than any of the other mice. Considering that B^0^AT1 (SLC6A19) is the major transporter of neutral amino acids at the intestinal epithelium and absorbs the bulk of diet-derived neutral amino acids from the intestinal lumen [17], it is highly probable that post-prandial amino acid level rises were curtailed in the NOD.*Slc6a19^−/−^* mice, consistent with the absence of higher plasma amino acids in NOD.*Slc6a19^−/−^* mice sampled. In C57Bl6J.*Slc6a19^−/−^* compared to C57Bl6J.*Slc6a19^+/+^* mice, we previously showed that plasma levels of neutral amino acids, including the BCAAs valine, isoleucine, and leucine, were significantly reduced in Slc6a19 deficient mice after a high protein meal, indicating that amino acid malabsorption is a characteristic of Slc6a19 deficient mice [19].

While the NOD mouse is an established model for type 1 diabetes, it does not accurately reflect all the characteristics of human type 1 diabetes [21]. It is becoming clearer that human type 1 diabetes is a heterogenous condition, whether it be by the presence or absence of overweight/obesity, genetic susceptibility factors, order of appearance of islet antibodies, severity of insulitis progression, and age at diagnosis [22,23]. Thus, it remains possible that SLC6A19 could become a therapeutic target for certain subgroups at high risk of type 1 diabetes, such as those who are overweight or have overlapping genetic risk for type 1 and type 2 diabetes.

In conclusion, Slc6a19 deficiency did not reduce the high incidence of insulitis and type 1 diabetes development in female NOD mice. The research focus on SLC6A19 as a potential therapeutic target in diabetes should be for type 2 diabetes for now [17,18], with the possibility of reconsidering it as a target for type 1 diabetes as we learn more.

## 4. Materials and Methods

### 4.1. Mouse Model

NOD/ShiLtJArc mice (referred to as NOD mice herein) were sourced from the Animal Resource Centre (Canning Vale, WA, Australia). All mice were housed in a temperature and humidity controlled environment on a 12 h light/dark cycle. NOD mice globally deficient in a functional *Slc6a19* gene were generated using the CRISPR-Cas9 system and were genotyped using a PCR assay. Female NOD.*Slc6a19^+/+^* and NOD.*Slc6a19^−/−^* littermates were fed standard rodent chow diet (Gordon’s Specialty Stockfeed, NSW, Australia) ad libitum from 6 to 30 weeks of age or until diabetes diagnosis (plasma glucose ≥ 20 mmol/L for 2 consecutive days) at which time the mice were ethically culled. All mice experiments were approved by the Animal and Experimentation Ethics Committee of the Australian National University (protocols: A2017/25 and A2017/44).

### 4.2. Generation of CRISPR/Cas9 Edited Slc6a19 Deficient Mouse Strain

For SpCas9 editing, two single guide RNAs (sgRNAs) were designed, guide 1 targeting exon 1, (5′- CCTGGCCTAGAGGAGCGGATTCC -3′) and guide 2 targeting exon 2, (5′- CATCGGTCAGAGGCTACGCAAGG -3′) of Slc6a19. The methodology for single guide RNA synthesis, as gBlock gene fragments (Intergrated DNA Techonologies, Baulkham Hills, NSW, Australia), is as previously described [24].

NOD studs and recipient mice (Animal Resource Centre, Perth, WA, Australia) were maintained at the Australian Phenomics Facility under specific pathogen–free conditions. Seven-wk-old NOD females were superovulated then mated with NOD males. Fertilized zygotes were collected from the oviduct and maintained under M16 medium (M7292; Sigma-Aldrich, St. Louis, MO, USA) overlaid with mineral oil. A mix of 50 ng/µL of SpCas9 protein (PNA Bio, Thousand Oaks, CA, USA), *Slc6a19*-purified gBlocks plasmids (guide 1, 5 ng/µL and guide 2, 5 ng/µL) were co-injected into the mouse zygotes. Pronuclear injections were performed as described [24], and microinjected zygotes were cultured overnight at 37 °C in a 5% CO_2_ incubator and then surgically transferred at two-cell stage into the ampulla of NOD pseudo-pregnant females.

Eight mice were Sanger sequenced, of which three revealed *Slc6a19* gene allele indels as outlined in the Appendix A. The ASD707:Maggot:::F0#13 mouse was found to be wild type for both alleles targeted by guide 1 (exon 1), but had a 7bp deletion and 2bp insertion in one of two alleles targeted by guide 2 (exon 2) (wild type allele, 5′- CATCGGTCAGAGGCTACGCAAGG -3′; mutated allele 5′- CATCGGTCAG-------**TG**GCAAGG -3′). The ASD707:Maggot:::F0#13 mouse was chosen as the NOD *Slc6a19^+/−^* founder mouse for this work. Following back-crossing of the founder mouse to NOD for 5 generations, keeping the compound 7 bp deletion and 2 bp insertion heterozygous *Slc6a19* mutation, marked aminoaciduria was confirmed in the phenotype of *Slc6a19^−/−^* offspring. Genotyping of mice was performed using DNA extracted from ear punch samples.

### 4.3. Body Weight, Glucose Measurement and Amino Acid Analyses

Serial measurements of body weight and 9 AM fed-state tail vein plasma glucose, using a glucose meter (StatStrip Xpress^TM^, Nova Biomedical, Flintshire, United Kingdom), were performed four-weekly (weeks 6–18) and fortnightly (weeks 18–30), or up until diabetes diagnosis. Urine and plasma samples were collected from NOD.*Slc6a19^+/+^* and NOD.*Slc6a19^−/−^* mice between 6–8 weeks of age before the development of diabetes in any mice. Urine samples were standardised by osmolality. Amino acid analyses were performed using a gas chromatography mass spectroscopy (GC-MS) method, as previously described [19].

### 4.4. Pancreas Histology

All mice were euthanased by cervical dislocation at the end of the experimental period (upon the diagnosis of diabetes or at week 30). Pancreases were quickly excised and fixed in 10% neutral buffered formalin solution before embedding in paraffin using routine laboratory protocols. Histological analysis was conducted on 4 µm-thick sections stained with haematoxylin and eosin. Islets per whole pancreas section were counted and individually graded for insulitis using a method previously described with scores of 0 = no insulitis, 1 = peri-islet insulitis, 2 = intermediate insulitis, 3 = intra-islet insulitis and 4 = complete islet insulitis [25]. Analysis was performed by two blinded observers.

### 4.5. Statistical Analysis

Results are presented as mean ± SEM or median with interquartile range. Repeated measures two-way ANOVA and Bonferroni post hoc tests were used to compare differences in time course data between groups and replicates, respectively (unless otherwise stated). Comparisons between two groups were performed using the t-test for parametric analyses, with *p*-value Bonferroni adjustment for multiple comparisons, or the Mann–Whitney test for non-parametric analyses. Analyses were performed using GraphPad Prism (Version 9.2.0). *p* < 0.05 was considered statistically significant. Diabetes free survival was compared between NOD.*Slc6a19^+/+^* and NOD.*Slc6a19^−/−^* groups using the log-rank (Mantel-Cox) test. Thirty-two mice per group was estimated to provide 80% power to detect an increase in diabetes free survival from 50% to 85% with α = 5%.

## Figures and Tables

**Figure 1 metabolites-11-00665-f001:**
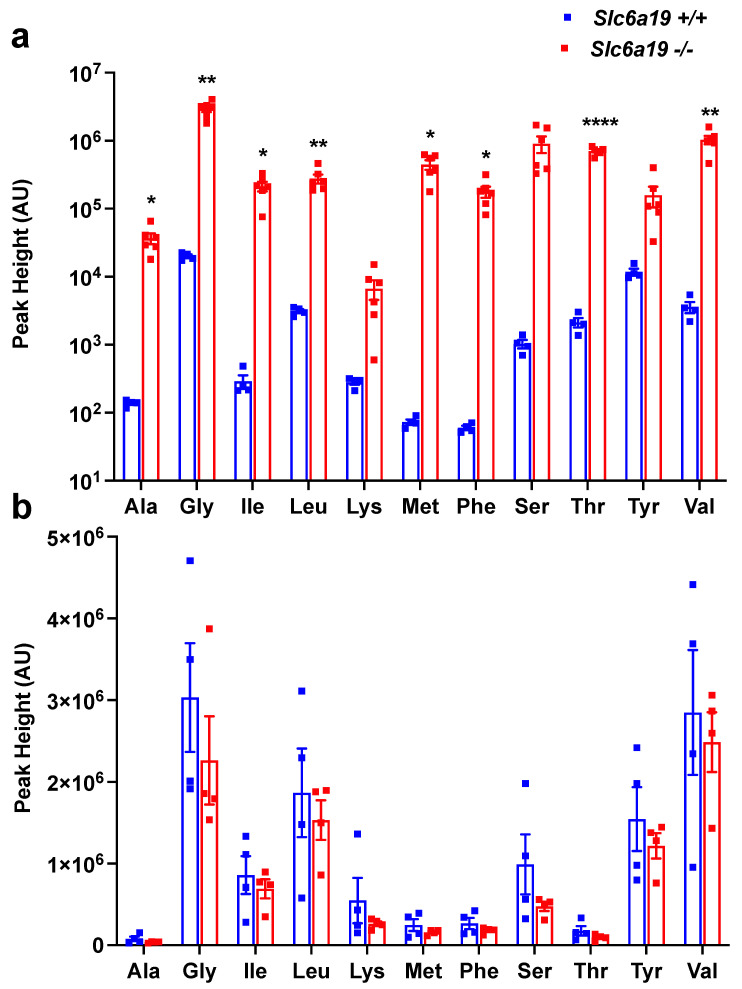
*Slc6a19* deficiency caused marked aminoaciduria in NOD mice. (**a**) Amino acid abundance measured in urine normalised to osmolality. (**b**) Plasma amino acid abundance. Samples were collected in the morning from mice 6–8 weeks of age, stored at −80 °C. Amino acid abundance is presented as peak height in arbitrary units (AU). All results are presented as mean ± SEM; * *p* < 0.05, ** *p* < 0.01, **** *p* < 0.0001 *Slc6a19^+/+^* versus *Slc6a19**^−/−^*; *n* = 4–6 mice/genotype.

**Figure 2 metabolites-11-00665-f002:**
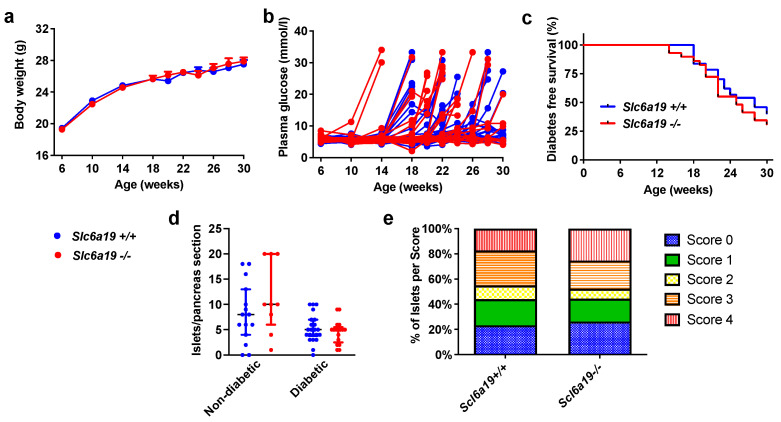
Slc6a19 deficiency in NOD mice does not alter body weight, incidence or timing of diabetes development, or severity of insulitis. (**a**) Body weight (g) and (**b**) fed-state plasma glucose (mmol/L) measured in the morning from weeks 6–30 of age, or until the time of diabetes diagnosis. Data are presented as mean ± SEM; *Slc6a19^+/+^*, *n* = 37; *Slc6a19^−/−^, n* = 29. (**c**) Diabetes free survival from weeks 6–30 of age. *Slc6a19^+/+^*, *n* = 37; *Slc6a19**^−/−^**, n* = 29; Mantel-Cox log rank test: *p* = 0.37. (**d**) Total number of islets per pancreas section according to genotype and diabetes status. *Slc6a19^+/+^*, *n* = 37 (*n* = 15 non-diabetic, *n* = 22 diabetic); *Slc6a19**^−/−^**, n* = 26 (*n* = 9 non-diabetic, *n* = 17 diabetic). (**e**) Number of islets at each grade of insulitis (grades 0–4) according to mouse genotype. *Slc6a19^+/+^*, *n* = 34; *Slc6a19**^−/−^**, n* = 26.

## Data Availability

All data is available in the Appendix A and on request to the corresponding author.

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
