# Peer review of "Knockout of the Amino Acid Transporter SLC6A19 and Autoimmune Diabetes Incidence in Female Non-Obese Diabetic (NOD) Mice"

_metabolites, 2021, doi:10.3390/metabo11100665_

Round 1
Reviewer 1 Report
The investigators postulate a relationship between plasma amino acid levels and susceptibility to type 1 diabetes. They model their hypothesis using NOD mice with a knockout of the amino acid transporter Slc6a19 resulting in aminoaciduria and deficiency of neutral amino acids. They find that homozygous knockout mice show incidence of diabetes by week 30 of 59.5% (22/37) versus 25 69.0% (20/29) in wild type NOD mice (hazard ratio 0.77, 95% 26 confidence interval 0.41-1.42; Mantel-Cox log rank test: P=0.37). They therefore conclude that Slc6a19 knockout does not affect susceptibility to diabetes in the NOD mouse. The procedures are excellent. Would point out that a hazard ratio of 0.77 with confidence interval 0.41-1.42 does not exclude their initial hypothesis. In order to confirm or deny their speculation, I calculate a sample size needed of 156 mice to achieve adequate power.
Author Response
Thank you for this feedback.
Our power calculations to show an improvement in survival rate without diabetes from 50% to 85% by 30 weeks in the ko mice was 64 mice (32/group). We agree that a larger number of mice would be needed to prove a smaller difference in survival. The non-significant trend, however, was in the opposite direction to our hypothesis, such that it is very unlikely that substantially increasing numbers would have resulted in support for our hypothesis.
Reviewer 2 Report
This paper from Waters et al neatly describes the finding that, against expectation, knocking out SLC6A19 in NOD mice did not reduce the incidence of autoimmune diabetes in this model. The design was straightforward with reasonable numbers of animals in the experimental and control groups. No additional comments.
Author Response
Thank you for your positive feedback.